# Development of a Sport Food Exchange List for Dietetic Practice in Sport Nutrition

**DOI:** 10.3390/nu12082403

**Published:** 2020-08-11

**Authors:** José Miguel Martínez-Sanz, Susana Menal-Puey, Isabel Sospedra, Giuseppe Russolillo, Aurora Norte, Iva Marques-Lopes

**Affiliations:** 1Research Group on Food and Nutrition (ALINUT), Nursing Department, Faculty of Health Sciences, University of Alicante, 03690 Alicante, Spain; josemiguel.ms@ua.es (J.M.M.-S.); isospedra@ua.es (I.S.); 2Unit of Nutrition and Dietetics, Faculty of Health Sciences and Sports, University of Zaragoza, 22002 Zaragoza, Spain; smenal@unizar.es (S.M.-P.); imarques@unizar.es (I.M.-L.); 3Spanish Academy of Nutrition and Dietetics (AEND), 31006 Navarra, Spain; g.russolillo@academianutricion.org

**Keywords:** food exchange list, sports foods, dietary supplements, dietetic practice, menu planning

## Abstract

Food exchange lists have been widely used in dietary practice in health and disease situations, but there are still no exchange lists for sports foods. The aim of this study was to apply a previous published methodology to design food exchange lists to the development of a sports food exchange list, with sport products available in Spain. A cross-sectional study of the nutritional composition of sports foods, regarding macronutrients and energy, was carried out. A total of 322 sports foods from 18 companies were selected, taking into account their interest in sports practice and with nutritional data provided by companies. Sports foods were divided into seven groups: sports drinks; sports gels; sports bars; sports confectionery; protein powders; protein bars; and liquid meals. A sports food composition database based on portion size usually consumed by athletes and/or recommended in commercial packaging was created. Within each sports foods group, different subgroups were defined due to differences in the main and/or secondary macronutrient. The definition of each exchange list with the amounts—in grams—of each sports food within each group and subgroup, was done using statistical criteria such as mean, standard deviation, coefficient of variation, and Z value. Final exchange values for energy and macronutrient have been established for each group and subgroup using a methodology to design food exchange lists previously published by the authors. In addition, those products with high Z values that can provide greater variability in dietary planning were included. The usefulness of sport foods lists as well as the use of an exchange system in the dietary practice of sports nutrition is discussed, and examples of how to use them with athletes are presented. This first sport foods exchange list showed in this study, with commercial sports products available in Spain, can be a novel tool for dietetic practice and also can allow sport nutrition professionals to develop another sport food list using the methodology described in this paper. Its management would allow dietitians to adapt dietary plans more precisely to the training and/or competition of the athlete.

## 1. Introduction

Sport is defined as physical activity, which requires good physical condition of participants and is related to athletic performance and health of athletes. Among these indicators, can be highlighted nutrition, diet, body composition, hydration, supplements, and doping [1].

The nutritional strategies used to meet the daily nutritional requirements for sport have been discussed in recent years [2]. A number of researchers, position/consensus statements, and scientific organizations have established recommendations regarding the intake of macro and micronutrients by athletes, according to the training/competition phase, the number of training sessions, and the periodization of the training [3,4,5]. There are also evidence-based protocols for the use of specific sports products—dietary supplements (DS)—in specific situations in sport when the nutritional intake is insufficient or inadequate [4,5,6]. DS are also known as ergonutritional aids, food supplements, sports food, or sports supplements [1]. According to the International Olympic Committee, DS were defined as “A food, food component, nutrient, or non-food compound that is purposefully ingested in addition to the habitually consumed diet to achieve specific health and/or performance benefit” [6]. More than 40% of athletes consume DS, making them the main target for the industry that produces DS [7,8,9]. Recently, Waller and colleagues found that the most widely used DS were sports drinks (70%), caffeine (48%), protein (42%), and sports bars (42%) [7].

The Australian Institute of Sport (AIS) created a DS classification system for athletes (known as the ABCD system) in which DS are differentiated according to the existing level of scientific evidence, as well as other parameters related to security, legality, and effectiveness in improving sports performance. This system is continually updated, being the last version of 2019 [8]. Thus, for the DS belonging to group A, there is a high level of scientific evidence for performance improvement [2,8]. According to the Academy of Nutrition and Dietetics about Nutrition and Athletic Performance in 2016 and AIS, the DS with group A evidence are sport foods [9], such as sports drinks, bars, and gels, or milk whey protein, (provide a source of nutrients when it is not practical to consume everyday food), medical supplements, and performance supplements [4].

The potential benefits of the use of DS include practical assistance to meet sports nutrition goals, the prevention or treatment of nutrient deficiencies, a direct ergogenic effect (in some cases), the reduction of gastrointestinal complaints and/or cramping; to help ease up digestion and prevent the gastrointestinal tract from permanent overload during special situations (e.g., by higher energy density or faster gastro-emptying) [9,10].

Knowledge of the protocols of use of DS and of the evidence regarding their effects is important for the dietetic and nutritional strategy of an athlete, for both training and competition. Sport nutrition professionals should base such interventions on nutritional goals, depending on the sports category, type of training/competition, intake of foods/liquids/DS, etc. [4,6]. Many of the DS used are sports foods, which are frequently included in the menu planning of training and competition days [11,12]. It is important to remember that athletes must test DS, timing and brands of DS in training sessions before a race/event where they can have detrimental effects as gastrointestinal problems [3,13].

The existence of DS lists with products of different brands but with a similar nutritional value between them, is highly useful in facilitating nutritional prescription. One of the greatest advantages of these lists is the fact that they can complement the athlete’s diet totally or partially and thus facilitate reaching their needs when it is difficult from a practical point of view to eat only food.

One of the methods used by dietitians for menu planning is the food exchange list system, a user-friendly tool that was developed to help individuals to adopt healthy eating habits and/or follow a specific diet plan [14].

Food exchange lists have been used for the last 70 years. The food exchange lists are groups of weighed foods that approximately contribute the same macronutrient value. Within each food list, one exchange is approximately equal to another in the three macronutrients (carbohydrates, proteins, and fats) and energy, and they can be exchanged in meal planning without significant differences in dietary intakes of patients [15,16]. The first edition was published in 1950 and was developed by the American Dietetic Association, the American Diabetes Association, and the United States Public Health Service [17]. Since that time, the American exchange lists have been updated several times [15,18,19] and used as a reference by many countries that have worked to design their own lists, to be used in the development of meal planning for healthy individuals or those with chronic disorders such as diabetes, obesity, and cardiovascular and kidney diseases. The food exchange lists have been updated to introduce new foods or adapt them to specific countries or populations [16,20,21,22]. In Spain, the authors have developed and validated food exchange lists, arranged according to the three macronutrients and energy [23] for general use as well as micronutrients and other nutrients of concern [24], to be used in different physiological and disease situations across the lifecycle, including vegetarians and vegans [25,26]. Currently, as far as we know, for athletes, there are no food exchange lists of specific sport foods that include the DS with scientific backing [4]. The interest of this study is that the authors apply a previous published methodology [23,24] to design a food exchange list of DS using sport products from well-known commercial brand in Spain but also in other countries. For one hand, the lists published in this work can be used as a tool for dietary practice in sports nutrition, and on another hand, the methodological approach described in this study can be applied by other researchers to develop new lists with other products adapted to other countries or regions. This study aimed to describe a methodological approach to design DS sport food exchange list and to develop a DS exchange list using DS and sports foods from the Spanish market. This unique and novel dietary tool can contribute to dietary planning in sports nutrition, facilitating the nutritional intervention and menu planning during training and/or competition.

## 2. Material and Methods

A cross-sectional study of sports foods and sports products was carried out of the most representative brands, regarding their nutritional composition of macronutrients and some micronutrients of interest for training and competition. The phases of this study are described below:

### 2.1. Selection of Nutrients and Sports Foods

The following key nutrients were selected: carbohydrates, sugars, proteins, and sodium, according to evidence-based sport dietetic practice over recent years [4,5,6,27]. The nutrients selected in this study are considered as evidence A level by the American Academy of Nutrition and Dietetics, International Society for Sports Nutrition, International Olympic Committee, and AIS [4,5,6,8]. The sports products from different national and international commercial brands with these nutrients of special relevance in sports nutrition were included [4,5,6,8].

The nutritional composition of the products included in this study was directly sent by the trademarks to the authors. The following sport products were excluded: DS whose nutritional composition does not have the nutrients previously considered or whose suppliers/manufacturers did not directly provide the necessary information on their nutritional composition. A database with the sports foods composition per portion size usually consumed by athletes and/or commercial container was created. After a general analysis of the nutritional composition, a total of 322 sports foods, from 18 different national and international companies, were selected and divided into seven groups according to the type of sports foods: (1) sports drinks; (2) sports gels; (3) sports bars; (4) sports confectionery; (5) protein powders; (6) protein bars; and (7) liquid meals (mixed macronutrient supplement).

### 2.2. Sports Foods Classification

Within each type of sport food, classification of DS in subgroups according to the main macronutrient, as well as an identification of food groups with a high content of some nutrients that should be taken into account in certain sports was also done. In this way, the products were classified into subgroups according to their predominant macronutrient (carbohydrates, sugars, proteins and a mix of carbohydrates, and proteins and a significant amount of each of the others). In accordance with the European Regulation on nutrition-related declarations for foods [1], the criteria for the selection of the significant nutrients in a food group were as follows: a food group was considered high in sugars when sugars comprised more than 5% of the mean net weight in solid foods (e.g., sport bars) and more than 2.5% in liquid foods (e.g., sport drinks). In the same way, the protein content was considered high when it provided at least 20% of the energy value of the product (e.g., protein bars). Following these criteria, the sports products were described as follows: products rich in carbohydrates (sports gels, sports confectionery, and sports bars), containing primarily carbohydrates and sugars; products rich in carbohydrates and electrolytes solutions (sports drinks), providing mainly sugars; protein-rich products (protein bars and protein powders), providing mainly proteins; and recovery products with carbohydrates and proteins (liquid foods and bars), providing mainly carbohydrates and proteins as shown in Table 1.

The control of the nutritional data in each sports foods exchange list was carried out using the statistical methods previously published by authors for the design of food exchange lists [23,24], described below.

Once the groups and subgroups were made, statistical criteria were applied to control that each food corresponded to its group and that the macronutrients values were similar between them according to the definition of an exchange food list.

The statistical parameters that control the homogeneity of foods within each list and that were calculated within each group were: the mean, standard deviation (SD), coefficient of variation (CV), and Z score values for energy and macronutrients (protein, carbohydrates including sugars, and fat). To validate the allocation of foods within an exchange list, the recommended values of SD used by Wheeler and colleagues [19] in exchange lists for energy and macronutrients (energy: 20 kcal, carbohydrates: 5 g, fat: 2 g, and protein: 3 g) were considered. If the SD values exceeded the limits, the amounts of foods were modified or the food was removed from the list and placed in another list where its macronutrient content would be appropriate. The foods and sport products that satisfied the SD statistical criterion for all the macronutrients were incorporated into the exchange lists. After this, the CV was also analyzed, aiming for values of less than 30% for energy and at least the main macronutrient. For groups with higher values of the CV, the Z value for each food was calculated: Z values between −2 and +2 were established as a criterion to exclude foods with high variations from a list. The statistical criteria proposed for the inclusion of sports foods within each exchange lists are shown in Table 2.

In all cases, the amounts in grams tested were based on common culinary measures and the recommendations on commercial containers of the sports products intended to facilitate dietetic practices (e.g., 2 teaspoons), or on the usual portion sizes (e.g., sports drinks: 1 serving of 500 mL), or on commercial containers/portions (e.g., 1 small bar of 30 g or an energy gel, 1 dose/serving of 20 g).

The determination of the final energy and macronutrient values assigned to each food exchange group and subgroup was performed according to the established rounding criteria previously published by the authors [23]. Values were rounded down for decimals less than 0.49 and rounded up for decimals higher than or equal to 0.50, as long as the Z value of the exchange group value against the mean group value was between ± 1. If the Z value was outside the limits, a new rounding was performed so that the established criteria were met. The energy value of each exchange group was calculated by multiplying the mean contents of proteins, fats, and carbohydrates assigned to each exchange list by the respective Atwater factors.

Due to the differences in the sodium content within each sports foods exchange group, the sodium content was also included so that the contribution of each group and subgroup to the resulting menu could be monitored.

## 3. Results

After the application of statistical parameters, the mean energy and macronutrient values for each group and subgroup were obtained as well as the the CV and SD, which are shown in Table 3. This table is the result of the application of the statistical criteria of SD of energy: 20 kcal, carbohydrates: 5 g, fat: 2 g, and protein: 3 g and CV less than 30% at least for the most important macronutrients within each type of sports food group and food group. Those foods that did not allow maintaining these criteria of CV and DS were withdrawn or their quantity modified, but maintained a portion of consumption commonly used by athletes.

In some types of sports foods such as sports bars, protein bars, protein powders, and liquid meals, they have been divided into several subgroups due to their nutritional composition and therefore to be able to meet the statistical criteria of DS, CV, and Z value within each group. Likewise, each subgroup also has an assigned value of macronutrients and energy in such a way that the foods included within each subgroup show greater homogeneity between them and therefore they can be exchanged between them maintaining a similar value of energy and macronutrients. This homogeneity within each group or subgroup achieved with the application of the statistical criteria, allows improving the nutritional precision in the prescription of these products, and at the same time, to have a greater ability to choose between several DS.

Although the Z value for each sports food was not indicated, it is important to note that some foods had a Z value outside the limits (Table 4). These sports foods should not be interchanged very frequently because that would modify the dietary intakes of athletes. These sports foods should be taken into account by professionals in menu planning in order to avoid high nutritional deviations from the mean values. If none of the foods listed in a group with a high CV showed Z values outside the limits, the variability of the group was disregarded.

The definite macronutrient values assigned to each sports foods exchange list after subjecting them to the rounding criterion previously exposed in the methods section and the energy value calculated by the Atwater system are shown in Table 5. There is a mean rounded value assigned to each group or subgroup. In sports foods rich in carbohydrates—such as sports drinks, gels, and bars—the carbohydrates per portion ranged from 24 to 35 g. These products had very low amounts of protein and fat (0 to 2 g per serving), except for subgroup 3 of the bars—which had 5 g of protein and 6 g of fat per serving and whose intake could be useful at a given time depending on the sporting activity of the athlete. The sports confectionary was also high in carbohydrates, providing 5 g per serving and 0 g of fat and protein.

In protein-rich products, such as protein powders and bars, the protein ranged from 11 to 26 g per serving. The amounts of the protein powders per exchange provided very little carbohydrates (2 g per exchange) and fat (0.5–1 g per exchange). Protein bars (with more than 20% of the caloric value in proteins) provided low amounts of fat (5 g per exchange) and moderate amounts of carbohydrates (14 and 18 g in Subgroups 1 and 2, respectively). Finally, the mixed group with carbohydrates and proteins (liquid meals) was divided into two subgroups, the first with a higher contribution of proteins and the second with a higher contribution of carbohydrates.

The sports foods exchange lists are shown attached to this manuscript (Appendix A). The exchange lists of the different types of sports foods and subgroups are showed in detail and as can be seen, each type of sports food has a list of products, in which they can be exchanged between them with a common value of energy and macronutrients. Some groups are a single list such as gels or sports drinks and other lists are made up of different subgroups due to their composition in carbohydrates, protein, or both. As expected, including several sport foods in a list will increase the possibilities of choosing products for the same dietary prescription in sport, keeping a constant value of energy and macronutrients.

## 4. Discussion

Our study aimed to describe a methodological approach to design a sport food exchange list, using previous published criteria to develop food exchange lists. The objective of work was specifically to develop a sport food exchange list using data of sport foods available in the Spanish market including national and international sport food trademarks. This work, on the one hand, exposes the methodology to design sport foods exchange lists, and on the other hand, presents the Spanish sport food exchange lists. This study allows other nutrition professionals to design sports foods exchange lists of each country using their own available food trademarks. In addition, these specific food exchange lists are a unique and novel tool for dietetic practice as a complement to meal planning, since it includes more than 300 DS from several companies, allowing the selection of different sport foods within the same group or subgroup with similar macronutrient characteristics. This food exchange list for sports was developed based on typical DS available for their nutritional prescription to athletes in Spain. Nevertheless, most of the commercial sports products included can be used at an international level, since they are available in other countries. All the amounts of the sports foods proposed have been established according to the size of the commercial containers or the serving dosage indicated by the manufacturer.

Within a food exchange group, one food exchange is approximately equal to another in terms of energy, carbohydrates, proteins, and fats, and it can be exchanged for any other food in the same list. Likewise, this sports foods exchange list was divided into different DS exchange groups (sports gels, bars, drinks, etc.); however, some groups were divided into subgroups based on the nutrient contents (e.g., different amounts of sodium in sports drinks or protein in sports bars for example). In this sense, the nutrition and dietetics practitioner could choose different subgroups according to the athlete’s nutritional goals [4,5]. As previously mentioned, these products are considered as “sports foods”, which can be defined as “specialized products used to provide a convenient source of nutrients when it is impractical to consume everyday foods” [4,8].

This sports foods exchange list allows estimation of the number of exchanges needed to meet the energy and macronutrients requirements of an athlete, as well as the appropriate foods in the daily menu for training and competition sessions. At the same time, it can be used to prevent nutrition-related adverse outcomes such as dehydration, hyponatremia, and gastrointestinal problems [1,4,28]. It is important to note that these lists can be used both as a complement to a menu with current foods or as a substitute for them depending on the athlete’s moment. In this way, the sport foods exchange list is a system to dietetic practice that allows flexibility in dietary prescription choosing different DS that is, the choice of different SD within the same group or subgroup with similar energy values and macronutrients while keeping dietary precision [23]. This exchange list complements and does not replace the meal plans: since the greatest application is in before, during, and after training/competition, where DS are mainly used. In this way, athletes can meet with the dietary-nutritional recommendations of carbohydrates, sodium, and liquids per hour during training sessions or competition periods. The dietary plan through sports foods exchange lists could contribute to augment athletes’ exercise capacity and performance [4,29].

In different countries or regions, there are usually food composition tables that collect nutritional information per 100 g of food [30]. In the same way, there are food databases for many types of sports DS [31,32], but there are no databases that collect sports DS nutritional information according to the size of the commercial container or the serving dose. The sport food exchange list compiled here achieves this, thus facilitating the adaptation of diet plans. In this way, this list could help the transfer of nutritional recommendations for the diet plans of athletes, so aiding sports dietitians or nutrition professionals [29]. Sports nutrition professionals have access to a lot of scientific literature about the nutritional needs of athletes, but more dietary-practice tools are necessary. As an example, the software Core Nutrition Planning—that performs nutritional planning for endurance events—has been created, but for the use of specific supplements [33]. Another example of a dietary-practice tool is the sports food exchange lists proposed in this research.

The choice of a sports food subgroup in a food exchange list will depend on the dietary-nutritional needs before, during, or after training and/or competition. The sports drinks, sports gels, sports bars, and sports confectionery groups can be used before or during training and/or competition, while the protein powders, protein bars, and mixed macronutrient supplement groups can be used after training and/or competition [4,5,6].

The current nutritional recommendations for sport events (training and competition) establish that during physical activity, whilst an event is taking place, the average hourly intake should be 500 mL of liquid, 250–350 mg of sodium, and 30–90 g of carbohydrates [4,5]. The carbohydrates guideline can be made in the following way: the average intake of carbohydrates should be 30 g during the first hour of exercise, 60 g in the second hour, and 90 g in the third hour, up to an intake of 120 g of carbohydrates per hour [34]. The athlete’s nutritional needs during exercise could be met with the intake of water, sports drinks, sports gels, sports bars, and food. In addition, the post-exercise nutritional recovery is contemplated through the intake of carbohydrates and proteins (0.8 g carbohydrate/kg body weight plus 0.2–0.4 g protein/kg body weight) [4]. In the two situations mentioned above, the sports foods included in our lists can be used.

Continuing with the previously mentioned, the carbohydrate needs during training or competition can be achieved using the following examples of food exchange lists: (1) one exchange of sports drink provides 33 g of carbohydrates; (2) one exchange of sports drink plus one exchange of sports gel provides 58 g of carbohydrates; (3) one exchange of sports drink plus one exchange of sports gel and one exchange of sports bar (Group 2) provides 93 g of carbohydrates; as well, three exchanges of sports gel diluted in 500 mL of water plus three exchanges of sports confectionery also provides 90 g of carbohydrates. In these examples, the recommended sodium and liquid intakes are also met. In addition, an example of a recovery option for an athlete with a body weight of 70 kg would be two exchanges of mixed macronutrient supplements. Table 6 shows several examples of dietary plans through this system. As with all foods, the gut training is necessary so that athletes can tolerate the recommended intake of carbohydrates, sodium, and liquids during training sessions or competition periods, as well as to decrease gastrointestinal problems [13].

The use of these sport foods exchange lists allows the sports dietitian or nutrition professionals to adjust more precisely the dietary plans developed and adapted to the training and/or competition of the athlete. This improves nutritional accuracy.

Thus far, different food exchange lists from different countries [14,23,35] or regions [36] have been published, adapted to the general population or disease situations such as diabetes [19] or kidney disease [22,37]. Other tools for dietary planning have also been developed [38], but none incorporate supplements to be applied in sport. This study has described a methodological way to design sport food exchange lists and also published the first sport food exchange lists with sport foods available in Spain.

In this way, this study contributes so that nutritional professionals can use these lists and this system or even make new lists with the foods available and consumed by athletes in their country or region, creating a useful and interesting tool for dietary practice. Authors recommend strict control of the selection of foods with Z-values outside the limits (Table 4) and do not recommend using a fraction of the exchange amount indicated to adjust the recommendations of the menu planning.

The main limitation of this study is the restricted number of DS brands and products included. The sports foods exchange list needs a periodic updating due to the innovation, improvement, and design of new DS by the companies and brands. This system complements (never replaces) the meal plans made by dietitians or nutrition professionals. In addition, the use of these sports lists by athletes should be tested, regarding the ease and utility of their use in: the planning of menus by the sport nutrition professional, the understanding of the diet plan by the athlete, and his/her adherence to the diet plan in different sports situations. In this first phase, this could not be carried out but it will be in the next phase. Indeed, future research could evaluate the effectiveness of the lists and the impact of the system among nutrition and dietetics practitioners and athletes.

## 5. Conclusions

This work exposes a methodology to develop a sports food exchange list adapted to DS marketed in each country or region. In addition, this study publishes the first sports foods exchange list developed, made up of commercial sports products available in Spain. This is a novel tool that could be very useful for nutrition professionals in the dietary-nutritional planning for athletes before, during, or after training and/or competition. The management of these food exchange lists could allow adjusting the dietary plans more precisely to the training and/or competition of the athlete, thus improving the precision of the nutritional prescription.

It is necessary to note that DS listed in this database are from different national and international companies, but it is possible that other brands present different nutritional profiles. Therefore, the authors recommend the use of nutrition facts from DS labels to check for possible deviation from the groups, and to select those that meet the methodology indicated in this work.

## Figures and Tables

**Table 1 nutrients-12-02403-t001:** Groups and subgroups of sport foods included, and criteria to select the net weight of the portions tested to define the sports foods exchange.

Sports Foods	Criteria	Sports Foods and Companies Included
**Sports Drinks**	Weight range: 35–55 g (one, two, or three servings in 400–500 mL). Sodium range: <250 mg and >250 mg.	Crown Sport Nutrition, Geo Nutrición, Infisport, Recuperat-Ión, Enervit, Nutrixxion, 226ERS, Victory Endurance, Weider, Etixx, Coca-Cola, Keepgoing, Nutrinovex
**Sports Gels**	Weight range: 25–60 g (one or two servings). Sodium range: <100 mg, 100–200 mg, and >200 mg.	Biofrutal Sport, Enervit, Victory Endurance, Maurten, Nutrixxion, GlucoUp!, Myprotein, Crown Sport Nutrition, 226ERS, Infisport, Recuperat-Ion, Etixx, Keepgoing, Nutrinovex
**Sports Bars**	Weight range: 25–60 g (one or two servings). Subgroup 1. <25 g carbohydrate. Subgroup 2. >30 g carbohydrate, 2 g of fat, and 2 g of protein. Subgroup 3. >30 g carbohydrate, 6 g of fat, and 5 g of protein.	Nutrinovex, Infisport, Geo, Etixx, Keepgoing, 226ERS, Nutrixxion, Victory endurance, Crown Sport Nutrition, Enervit
**Sports Confectionery**	Weight range: 4–8 g (one, two, or four servings).	GlucoUp!, 226ERS, Victory endurance
**Protein Powders**	Weight range: 25–40 g (one, two, or three servings). Subgroup 1. Between 20 and <25 g protein. Subgroup 2. Between ≥25 and 30 g protein.	Weider, Myprotein, Crown Sport Nutrition, 226ERS, Enervit, Victory endurance, Keepgoing, Etixx, Infisport, Nutrixxion.
**Protein Bars**	Weight range: 30–80 g (one or two servings) Protein content >20%. Subgroup 1. <15 g protein per serving. Subgroup 2. >20 g protein per serving.	Victory Endurance, Weider, 226ERS, Nutrixxion, Enervit, Infisport, Myprotin, Keepgoing, Etixx, Paleobull,
**Liquid Meals**	Weight range: 30–60 g (one, two, three, or four servings) Subgroup 1. Ratio of 1 Carbohydrate/1 protein. Subgroup 2. Ratio of 2–4 Carbohydrate/1 protein.	Infisport, 226ERS, Etixx, Crown Sport Nutrition, Geo, My protein, Weider, Keepgoing, Enervit, Recuperat-Ion

Design of the exchange list for each type of sport food.

**Table 2 nutrients-12-02403-t002:** Criteria proposed for the definition of the sports foods exchanges in each group. Definitive values of macronutrients and energy assigned to each food sport exchange list.

Macronutrient	Standard Deviation (SD) for Each Group	Coefficient of Variation for Each Group (CV)	Z Values for Each Food
Energy	20 kcal	30%	2
Carbohydrate	5g		
Fat	2g		
Protein	3g		

**Table 3 nutrients-12-02403-t003:** Mean, standard deviation, and coefficient of variation of energy and macronutrients values of the sports foods groups and subgroups proposed in the exchange list.

Sports Food Exchange Groups	*N*	Energy kcal (SD)	CV (%)	Protein g (SD)	CV (%)	Fats g (SD)	CV (%)	Carbohydrate g (SD)	CV (%)
**Sports Drinks**									
**Sports Drinks**	**46**	137.25 (19.94)	14.53	1.32 (1.91)	144.96	0.03 (0.16)	481.04	32.95 (3.75)	11.38
**Sports Gels**									
**Sports Gels**	**71**	101.28 (17.35)	17.14	0.16 (0.28)	175.73	0.02 (0.05)	249.30	25.12 (4.25)	16.93
**Sports Bars**									
**Sports Bars. Subgroup 1**	**17**	117 (21.06)	18	1.27 (1.13)	89.12	1.86 (1.81)	97.21	23.8 (3.68)	15.45
**Sports Bars. Subgroup 2**	**24**	167.5 (33.3)	19.9	2.2 (1.4)	64.7	2.2(1.7	76.3	34.6 (4.5)	13
**Sports Bars. Subgroup 3**	**17**	203.9 (21.4)	10.5	4.6 (2.3)	49	6.0 (1.5)	24.8	33.0 (4.4)	13.4
**Sports Confectionery**									
**Sports Confectionery**	**9**	19.86 (5.75)	28.95	0.07 (0.12)	165.30	0.11 (0.16)	142.26	4.64 (1.29)	27.88
**Protein Powders**									
**Protein Powders. Subgroup 1**	**50**	108.72 (9.74)	8.95	22.64 (1.16)	5.11	1.11 (0.67)	60.34	2.1 (0.94)	45.03
**Protein Powders. Subgroup 2**	**20**	116.01 (10.58)	9.12	25.82 (1.0)	3.86	0.7 (0.76)	109.29	1.62 (1.05)	65.19
**Protein Bars**									
**Protein Bars. Subgroup 1**	**27**	158.42 (25.60)	16.16	10.86 (1.57)	14.5	4.81 (1.76)	36.55	17.93 (4.38)	24.42
**Protein Bars. Subgroup 2**	**16**	191.76 (57.69)	0	22.68 (30.8)	13.59	5.19 (1.23)	23.63	16.59 (3.88)	28.57
**Mixed Macronutrient Supplements**									
**Liquid Meals. Subgroup 1**	**8**	176.02 (14.5)	8.24	20.67 (1.3)	6.3	2.25 (0.59)	26.31	18.28 (1.47)	8.04
**Liquid Meals. Subgroup 2**	**17**	154.58 (18.09)	11.7	9.74 (1.74)	17.83	0.35 (0.39)	111.56	28.11 (3.85)	13.71

SD: standard deviation; CV: coefficient of variation.

**Table 4 nutrients-12-02403-t004:** Foods with Z values outside the limits (± 2) for the main macronutrients of the sports foods groups.

Sports Food Exchange Groups	Macronutrient	Sports Foods with Z <−2	Sports Foods with Z >2
**Sports Drinks**			
**Sports Drinks**	Carbohydrate	Energy carbo charge—chocolate (Keepgoing)	Powerade^®®^ Ice Storm (Cocacola), Powerade^®®^ Blood Orange (Cocacola)
**Sports Gels**			
**Sports Gels**	Carbohydrate	-	Longovit Gel—strawberry and banana (Nutrinovex)
**Sports Bars**			
**Sports Bars. Group 1**	Carbohydrate	-	-
**Sports Bars. Group 2**	Carbohydrate	Enervit power sport competition—orange (Enervit)	-
**Sports Bars. Group 3**	Carbohydrate	-	Protein bars—chocolate (Enervit)
**Sports Confectionery**			
**Sports Confectionery**	Carbohydrate	-	-
**Protein Powders**			
**Protein Powders. Subgroup 1**	Protein	Day & night casein chocolate (Weider), Isolate protein drink—chocolate (226ERS), Sequential Protein—chocolate (Cronw Sport Nutrition), Impact Whey protein (My Protein), The whey + (My Protein).	-
**Protein Powders. Subgroup 2**	Protein	-	Sequential protein—chocolate (Infisport), K-weeks immune—chocolate (226ERS)
**Protein Bars**			
**Protein Bars. Subgroup 1**	Protein	-	Vegan protein bar—berries (Victory Endurance), Enervit power sport—crunchy cookie (Enervit)
**Protein Bar. Subgroup 2**	Protein	-	-
**Mixed Macronutrient Supplements**			
**Liquid Meals. Subgroup 1**	Carbohydrate and protein	Top 50/50 recovery 1:1 leucina+—chocolate (Infisport)	-
**Liquid Meals. Subgroup 2**	Carbohydrate and protein	-	Recovery shake chocolate (Etixx)

**Table 5 nutrients-12-02403-t005:** Definitive exchange values of energy, macronutrients, and the Z value of the groups and subgroups proposed in the sports foods exchange list.

Sports Food Exchange Groups	Energy (kcal)	Z	Carbohydrate (g)	Z	Protein (g)	Z	Fat (g)	Z
**Sports Drinks**								
**Sports Drinks**	138	0.04	33	0.01	1	−0.17	0	−0.21
**Sports Gels**								
**Sports Gels**	101	−0.02	25	−0.03	0	−0.57	0	−0.40
**Sports Bars**								
**Sports Bars. Subgroup 1**	118	0.05	24	0.01	1	−0.24	2	0.08
**Sports Bars. Subgroup 2**	166	−0.04	35	0.08	2	−0.15	2	−0.15
**Sports Bars. Subgroup 3**	206	0.10	33	0.0	5	0.2	6	0.0
**Sports Confectionery**								
**Sports Confectionery**	20	0.02	5	0.28	0	−0.6	0	−0.7
**Protein Powders**								
**Protein Powders. Subgroup 1**	109	0.03	2	−0.1	23	0.31	1	−0.16
**Protein Powders. Subgroup 2**	119	0.28	2	0.36	26	0.18	0.5	−0.26
**Protein Bars**								
**Protein Bars. Subgroup 1**	161	0.10	18	0.02	11	0.09	5	0.11
**Protein Bars. Subgroup 2**	192	0.0	14	0.1	23	0.10	5	−0.15
**Mixed Macronutrient Supplements**								
**Liquid Meals. Subgroup 1**	176	0.0	18	−0.19	21	0.26	2	−0.42
**Liquid Meals. Subgroup 2**	154.5	0.0	28	−0.03	10	0.15	0.5	0.38

**Table 6 nutrients-12-02403-t006:** Examples of dietary plans through sports foods exchange list.

**Case Description**		
Event: football match. Duration: Two halves of 45 min long each. Between the two halves, there is a break interval of 15 min approximately.	Event: 63 km mountain bike race. Duration: 3 h. Mountain bike rider of 78 kg body weight.	Event: post-exercise recovery. Athlete of 60 kg body weight
**Dietary-Nutritional Recommendations ***		
CHO: 30 g/h Liquid: 500 mL/h	CHO: 30–60 g/h Liquid: 500 mL/h	CHO: 0.8 g/kg body weight (48 g) Protein: 0.2–0.4 g/kg body weight (12–24 g)
**Dietary Plan Based on Sports Exchange List**		
1 I sports drinks or 1 I sports gels + 500 mL water during break Interval	First hour: 2 I sports gels + 500 mL water in bike bottle. Second hour: 2 I sports gels + 500 mL water in bike bottle. Third hour: 2 I sports drinks.	1 I protein bar—subgroup 2 + 1 I sports bar—subgroup 2 + 500 mL water

CHO: carbohydrates; I: Exchange. * Dietary-nutritional recommendations based on American Academy of Nutrition and Dietetics, International Society for Sports Nutrition, International Olympic Committee, and Australian Institute of Sport [4,5,6,8].

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
