# Peer review of "Development of a Sport Food Exchange List for Dietetic Practice in Sport Nutrition"

_nutrients, 2020, doi:10.3390/nu12082403_

Round 1
Reviewer 1 Report
I am doubting whether this manuscript is of scientific relevance. When evaluated against key scientific criteria, it seems to fall short on e.g., verifiability and abstractness. Especially the latter is of concern. Science proceeds on a plane of abstraction. A general scientific principle is highly abstract. It is not interested in giving a realistic picture. While the proposed exchange list could be relevant for dietetic practice, it is unclear how this works contributes to the prevailing theoretical frameworks. This should be better reflected both in the abstract as well as throughout the paper to guide the readership of this journal on how this is relevant to them. Overall, I think substantial changes should be made to improve this manuscript. The authors should not just make the changes suggested, but also improve the overall quality of the work.
Some overall points to consider:
- Write in the past tense when evidence from the past is discussed. Reserve the present tense for matters of fact or matters that are well accepted and will be considered as 'true' at present and likely into the future present, too.
- Throughout the body of your paper, whenever you refer to outside sources of information, or to an idea or piece of information that you learnt from a text, you must cite the sources from which you drew that information.
- For the introduction section, the readability could be improved by removing unnecessary details and revising the structure. Generally, start with the overall scope of the topic and clearly identify the problem that you will focus on in this manuscript. Give a clear indication of the theoretical framework you use, and the gap in the literature that your research will fill. Lastly, concretely list the contributions that your paper will make to the state-of-the-art in the field.
- For the discussion section, consider your study's limitations better and focus on both the theoretical and practical (public health/legislation) implications of your work.
- It would be interesting to discuss also how useful it is for athletes to use these tables for menu planning and by extension to regulate properly their CHO intake. Moreover, it would be good to address whether food exchange lists could be part of dietetic manipulations to augment athletes’ exercise capacity and performance
- Although my comments are not exhaustive, please see below a number of points that may help to improve the manuscript.
Abstract and title:
L15-16: Can you provide rationale why these lists have not been developed for sport foods? Also, why are the currently available food exchange lists not suitable for athletes?
L16-17: Unclear what you mean with “previously published statistical criteria”. Can you illuminate with an example?
L18: How were the foods selected?
L26: What do you mean here? Please clarify
L27: How relevant is this across the globe? It seems that these foods are only available in selected geographies. Please list this clearly.
The abstract is not coherent with the rest of the manuscript. Clearly define the topic and methodology, and why this methodology was used (and how it was conducted). Also note the concrete conclusions of your research.
Introduction:
L34-35: Please rewrite this first sentence, it is hard to read.
L48: Protein powder? Or protein bars?
L67: As suggested in my general remarks, please restructure the introduction section. Food exchange list systems are an integral part of your argument; suggest to define them clearer – perhaps using also an illustrative figure/table.
L69: Please explain to the readership how food exchange lists can have different editions
L78: This is not a scientifically sound hypothesis. First, a hypothesis must state an expected relationship between variables. Second, it must be testable and falsifiable; researchers must be able to test whether a hypothesis is truth or false. Third, it should be consistent with the existing body of knowledge. Finally, it should be stated as simply and concisely as possible.
L79: Please summarize the results of this research; it might not be known to the readership
Materials and methods:
L88: Which studies? [4]?
L90: Which evidence?
L95-96: The four key nutrients you selected (CHO, Sugars (which is also a CHO(?)), proteins and sodium) are all mandatory on food labelling. Could this information to be used? How were the levels of these nutrients measured in the included products?
Results:
L165-169: Are these then sports products?
Author Response
Dear Reviewer,
We attached a cover letter to your comments.
Thanks to your comments, we have made important changes to the manuscript to improve its compression, usefulness and application.

Reviewer 2 Report
This paper prsents novel focus, and there seems much potential!
However, I cannot see until page 9 what is better with this tool compared to careful planning an individualized sports-race-training-related dietary strategy for different occasions. Maybe this shoudl be clearly elaborated, maybe choose 1 example to highlight in which concrete situations this is of additional benefit compared to usual sports-diet planning - to me the benefit of this exchange list is not clearly delivered, although I can feel the potential; with this, until page 9 to me not clear what is to be exchanged by another, as athletes travelling for races mostly have their food and supplements with them!

Author Response
This paper presents novel focus, and there seems much potential!
However, I cannot see until page 9 what is better with this tool compared to careful planning an individualized sports-race-training-related dietary strategy for different occasions. Maybe this should be clearly elaborated, maybe choose 1 example to highlight in which concrete situations this is of additional benefit compared to usual sports-diet planning - to me the benefit of this exchange list is not clearly delivered, although I can feel the potential; with this, until page 9 to me not clear what is to be exchanged by another, as athletes travelling for races mostly have their food and supplements with them!
AUTHOR'S COMMENTS: According to the reviewer’s suggestion, the text has been modified to include early in the manuscript information about the interest, usefulness, and specific objective of this research.
Specific remarks in pdf file.
AUTHOR'S COMMENTS: The reviewer's comments made in the pdf file have been answered in the same pdf file.

Round 2
Reviewer 1 Report
Dear authors,
Thank you very much for your detailed rebuttal letter and revised manuscript. Unfortunately, my doubts on the scientific relevance of this manuscript are still present. It still seems that the methodological approach as described in [23,24] is merely applied to create a sports foods DS exchange list. In my view, it does not add novel methodological considerations nor is it described with the level of abstractness necesarry to proceed scientific knowledge on the matter.
The manuscript also still requires extensive editing of English language and style.
Please also note that the usage of food labels as data points (or specifications provided by manufacturers) are questionable. In Europe, tolerances are specified - which might impact siginficantly the results of your exchange list (https://ec.europa.eu/food/sites/food/files/safety/docs/labelling_nutrition-vitamins_minerals-guidance_tolerances_1212_en.pdf).
Author Response
Thank you very much for your comments. In fact, when rereading your reflections on our work, we believe that it is important to publish the lists as an annex to the manuscript so that the importance, usefulness and novelty of the tool we have created can be seen better.
As mentioned above, the authors have developed a methodology for making food exchange lists and we have applied them to make lists of frequent foods for the population, lists with nutrients of interest for certain pathophysiological states and also lists for the vegetarian and vegan population.
This work is not intended to be a methodological innovation but to present a novel tool. They are the first lists in the world about sports products and we are convinced will help dietary planning in sports practice. The reviewer is right, if this manuscript intends to present a novel tool, this work only makes sense accompanied by the lists. The authors have considered useful and necessary to publish the lists in an annex attached to this manuscript and thus enhance the value and utility of this novel tool. We hope that many professionals can use them in their daily practice.
Finally, throughout all the text, some sentences and paragraphs have been rewritten again to improve the style of English. Also, the methodology about how the lists have been obtained has also been better explained. Results and discussion section have been also improved.
Reviewer 2 Report
The paper has much improved, congrats!
I feel only still necessary to highlight in addition to the justification at the end of introduction which is now good (and in Table 5, but needed to be introduced even earlier in the paper), WHY this exchange list is better/should be preferred by athletes and sports dietitians for competitive and travelling situations, as to me this seems the major advantage to be helpful and aid such situations.
Author Response
Thank you very much for your comments.
The authors have taken note of your comment and the usefulness and advantage of these lists in the diet of athletes is named soon in the introduction.
We also inform you that the manuscript was rewritten in some parts in order to better understand the methodology and to improve the results and discussion section.
Finally, the authors have considered useful and necessary to publish the lists in an annex attached to this manuscript and thus enhance the value and utility of this novel tool.
We hope that many professionals can use them in their daily practice
Best regards